# Hallucination Mitigation in Large Vision-Language Models via Adaptive Multi-Subspace Projection

## Abstract

Recent advances in large vision-language models (LVLMs) have enabled powerful multimodal reasoning by integrating visual encoders with large language models (LLMs). However, their reliability is frequently undermined by hallucinations—generated text that inaccurately describes the visual input. Although fine-tuning can mitigate this, it is computationally expensive and demands large, curated datasets, making training-free alternatives more appealing. Among training-free strategies, model-editing is a more promising solution than decoding-based approaches. While decoding methods can adapt outputs per-input, they introduce substantial computational overhead and instability. Model-editing, by contrast, modifies the model's internal representations offline, offering a more efficient and stable framework. However, the effectiveness of current model-editing techniques is limited. Existing methods typically rely on a single, global subspace to correct errors. This static, one-size-fits-all approach treats all test samples identically, failing to capture the diverse modes of hallucination that vary from one input to another. To overcome this specific limitation, we propose a training-free hallucination mitigation framework that performs dynamic, per-instance suppression at test time. Our method advances the model-editing paradigm by first constructing a set of Disentangled Hallucination Subspaces, where each subspace isolates a distinct hallucination mode. Then, at inference, our model adaptively calculates weights to determine how a given input relates to each subspace. These weights guide a dynamically combined projection that selectively suppresses the most probable hallucination directions for that specific instance while preserving image-grounded semantics. Extensive experiments across multiple vision-language benchmarks and LVLMs families demonstrate consistent improvements, highlighting the robustness, generalizability, and efficiency of our approach.

## 1 Introduction

In recent years, integrating vision models with large language models (LLMs) has become a standard approach to leverage the advanced reasoning capabilities of LLMs. This integration has led to the emergence of large vision-language models (LVLMs) Liu et al. (2023a); Zhu et al. (2023). Despite rapid progress in this field, LVLMs remain unreliable in certain scenarios due to the persistent issue of hallucination, where the model generate irrelevant or non-factual content that is inconsistent with the input image.

Existing strategies for reducing hallucinations in LVLMs generally fall into two main categories: (i) fine-tuning approaches Xiao et al. (2025); Yu et al. (2024) and (ii) training-free methods Leng et al. (2024a); Wang et al. (2024); Yang et al. (2025). Although the fine-tuning approach often achieves superior performance, it requires curated datasets and significant computational resources, limiting its practicality in real-world deployments. Thus, training-free methods have become increasingly popular due to their flexibility and efficiency.

Existing training-free techniques can be broadly categorized into decoding-based Leng et al. (2024a); Wang et al. (2024) and model-editing Yang et al. (2025) approaches. Decoding-based methods, such as contrastive decoding Leng et al. (2024b), adapt model outputs per input during

Figure 1: **(Left)**: Existing model editing methods Yang et al. (2025) derive a single subspace of hallucination directions (via SVD) and apply fixed editing to the LVLM, using the same edited model for all inputs. **(Right)**: Our adaptive method identifies multiple subspaces from different hallucination modes and adaptively adjusts their contributions to edit the model based on the input image, enabling more flexible and context-aware hallucination suppression.

or after generation. Although effective, they incur substantial computational overhead due to additional forward passes and often suffer from instability. In contrast, model-editing approaches modify the internal layers of LVLMs offline, typically by projecting hidden representations into a low-dimensional subspace that captures hallucination-sensitive directions. However, prior methods rely on a single global subspace that is applied uniformly across all test samples, which limits their generalization capacity: the hallucination direction captured by this subspace may not align with the diverse hallucination patterns observed during inference. Specifically, existing editing approaches Yang et al. (2025) (see Figure 1 **(left)**) perform fixed model editing, intervening in the internal layers solely according to this global subspace, regardless of the input. This motivates the need for an adaptive strategy that can dynamically adjust at test time. To this end, we propose an adaptive model-editing framework (see Figure 1 **(right)**) that leverages multiple disentangled subspaces. Each subspace is better aligned with a particular hallucination mode, and their contributions to model editing are adaptively weighted based on the input sample itself. Overall, our framework operates in two steps:

**Step 1:** In a preprocessing stage, we construct multiple low-dimensional subspaces, each representing a distinct hallucination direction. Specifically, we leverage a paired dataset where each image is associated with both a hallucinated and a truthful description. For every image, we compute the state differences between hallucinated and truthful captions. These per-layer difference vectors are then clustered using K-means. Within each cluster, we apply Singular Value Decomposition (SVD) to extract an orthonormal basis that captures the dominant hallucination direction. This process yields a collection of subspaces, each characterizing a different hallucination-sensitive direction.

**Step 2:** At inference time, the precomputed subspaces from the offline stage are adaptively combined based on the given test sample. Specifically, we probe the LVLM with both the original image and a masked variant designed to induce hallucination. The difference between their hidden states provides an input-specific hallucination signal, capturing the model's susceptibility to hallucinated content. This signal is then projected onto the previously constructed subspaces, where the projection magnitudes act as relevance scores. These scores are used to compute adaptive weights for combining the corresponding subspace bases. The resulting weighted projection matrix is finally applied to the model's internal representations to suppress hallucinations during inference.

Unlike fixed editing methods Yang et al. (2025), our framework enables adaptive, fine-grained, and context-aware model editing, effectively filtering hallucinated content while preserving image-grounded semantics. In summary, our contributions are as follows:

- **Disentangled Hallucination Subspaces:** an offline construction of multiple low-rank subspaces, each capturing a distinct hallucination mode.

- **Adaptive Test-Time Mitigation:** a training-free framework that extracts an input-specific hallucination signal and projects it onto the precomputed subspaces. The resulting adaptive weighting dynamically suppresses hallucination directions during inference.

- **Extensive Evaluation:** We conduct experiments across multiple vision-language benchmarks and VLM families, demonstrating consistent and generalizable improvements across evaluation metrics.

## 2 RELATED WORK

**Large Vision-Language Models:** Large Vision-Language Models (LVLMs) have advanced rapidly, with designs such as BLIP-2 Li et al. (2023b), InstructBLIP Dai et al. (2023), MiniGPT-4 Zhu et al. (2023), LLaVA Liu et al. (2023b), mPLUG-Owl2 Ye et al. (2024), and Qwen-VL Bai et al. (2023) enabling strong multimodal capabilities. BLIP-2 Li et al. (2023b) employs a Query Transformer (Q-Former) to extract a fixed set of informative visual tokens from a frozen vision encoder, which are then passed to a frozen LLM—enabling modular alignment with minimal adaptation. InstructBLIP Dai et al. (2023) builds on BLIP-2 by applying instruction tuning across diverse vision-language tasks to improve generalization and alignment. However, bottlenecks such as the limited number of visual tokens and reliance on frozen backbones can lead to vision-to-language misalignment, contributing to hallucination risks. Linear projection approaches, such as MiniGPT-4 Zhu et al. (2023) and early LLaVA Liu et al. (2023b), preserve CLIP visual features but rely on weakly supervised tuning, which risks semantic misalignment during generation. mPLUG-Owl2 Ye et al. (2024) improves grounding through adaptive modality collaboration modules and broad instruction tuning across diverse tasks. Hallucinations—where models generate content ungrounded in the image—are exacerbated by incomplete grounding and the use of next-token training objectives that fail to penalize unfaithful outputs Li et al. (2023c). Early work in image captioning attributes hallucination to biased decoders and limited visual grounding, foreshadowing similar challenges in modern LVLMs. Subsequent evaluations Li et al. (2023c) emphasize that next-token training objectives further promote unfaithful outputs due to weak alignment constraints. Benchmark datasets such as CHAIR Rohrbach et al. (2018) and POPE Li et al. (2023c) have exposed persistent grounding failures. To address these issues, LLaVA-RLHF Sun et al. (2023) introduces factually-augmented reinforcement learning with human feedback (RLHF) and proposes MMHalBench, a benchmark specifically designed to assess hallucination in LVLMs, showing that targeted supervision can significantly improve factual alignment.

**Hallucination Mitigation Strategies:** Mitigation strategies for hallucination in vision-language models generally fall into four categories. (1) Null-space projection: Nullu Yang et al. (2025) suppresses hallucinations by projecting input features into the null space of a learned hallucination subspace (HalluSpace). While effective, it applies a single global HalluSpace and cannot adapt to sample-specific hallucination patterns. (2) Latent space steering: VTI Liu et al. (2024) applies fixed latent offsets to stabilize vision-language features during decoding. Although training-free, it lacks input adaptivity, as fixed shifts may be suboptimal across different scenes or prompts. (3) Token sparsification and contrastive decoding: VASparse Zhuang et al. (2025) filters low-importance visual tokens using attention sparsity and performs contrastive decoding between full and pruned tokens. While efficient, it does not semantically model hallucination features. (4) Constrained and contrastive decoding: Several methods fall into this category. HALC Chen et al. (2024) reweights decoding using adaptive visual context and contrast signals. DoLa Chuang et al. (2023) compares internal representations to detect and suppress hallucinated content. OPERA Huang et al. (2024) applies over-trust penalties and retrospection to discourage unsupported generations. VCD Leng et al. (2024a) performs contrastive decoding using perturbed visual inputs, and Woodpecker Yin et al. (2024) verifies and replaces hallucinated entities by cross-checking them with image-grounded evidence. Despite these advances, most existing methods are either global, heuristic-driven, or computationally intensive. In contrast, our method constructs multiple low-rank Disentangled Hallucination Spaces and performs sample-specific null-space projection at test time. It is entirely training-free, adapts to individual inputs, and avoids reranking or model retraining.

## 3 PRELIMINARY

**Vision-Language Alignment.** The input to a vision-language foundation model (LVLM) consists of an image $\mathbf{I}^{(i)} \in \mathbb{R}^{H \times W \times C}$ and a textual query $\mathbf{q}^{(i)}$. A vision encoder (e.g., ViT Dosovitskiy et al. (2021), CLIP Radford et al. (2021)) first extracts image features from $\mathbf{I}^{(i)}$. These features are then mapped into the language model's input space by a vision-language alignment module

(e.g., Q-Former Li et al. (2023a) or a linear projection), producing a sequence of $N$ visual tokens: $\mathbf{X}^{(i)} = [\mathbf{x}_0^{(i)}, \mathbf{x}_1^{(i)}, \ldots, \mathbf{x}_{N-1}^{(i)}], \quad \mathbf{x}_n^{(i)} \in \mathbb{R}^d$. Simultaneously, the textual query $\mathbf{q}^{(i)}$ is tokenized into a sequence of $M$ tokens: $\mathbf{T}^{(i)} = [\mathbf{t}_N^{(i)}, \mathbf{t}_{N+1}^{(i)}, \ldots, \mathbf{t}_{N+M-1}^{(i)}], \quad \mathbf{t}_m^{(i)} \in \mathbb{R}^d$. The combined input to the LVLM is the concatenated sequence $[\mathbf{X}^{(i)}, \mathbf{T}^{(i)}]$ of total length $J = N + M$.

**Model Forwarding.** The combined input sequence $[\mathbf{X}^{(i)}, \mathbf{T}^{(i)}] \in \mathbb{R}^{J \times d}$, where $J = N + M$, is then passed through the language model component of the LVLM. Let $L$ denote the total number of transformer layers, and $\mathbf{z}_{\ell,j}^{(i)} \in \mathbb{R}^d$ represent the hidden state corresponding to token index $j$ at layer $\ell$ for sample $i$. The model produces a sequence of contextualized embeddings:

$$\left\{ \mathbf{z}_{\ell,j}^{(i)} \right\}_{\ell=1, j=1}^{L, J} = f_\theta^{\text{LVLM}} \left( \mathbf{I}^{(i)}, \mathbf{q}^{(i)} \right). \tag{1}$$

These hidden states serve as the basis for downstream reasoning and generation tasks and are used in subsequent modules for hallucination suppression.

**Response Generation.** Following the forward pass through the LVLM, the final-layer hidden states $\{\mathbf{z}_{L,j}^{(i)}\}_{j=1}^J$ are used to generate the output response. Specifically, the model performs autoregressive decoding based on the attended multimodal context, producing the textual response token by token. The probability of the next token $y_{t+1}^{(i)}$ is modeled as:

$$P \left( y_{t+1}^{(i)} \mid y_{1:t}^{(i)}, \mathbf{z}_{L,1:J}^{(i)} \right) = \text{softmax} \left( \mathbf{W}_o \, \mathbf{h}_t^{(i)} \right), \tag{2}$$

where $\mathbf{h}_t^{(i)}$ is the decoder's hidden state at time $t$, and $\mathbf{W}_o \in \mathbb{R}^{V \times d}$ is the output projection matrix for a vocabulary of size $V$. Decoding continues until an end-of-sequence token is generated or a predefined maximum length is reached.

## 4 METHOD

LVLMs are highly prone to hallucination, generating textual outputs that are inconsistent with the input image. Existing training-free mitigation approaches employ either sample-agnostic preprocessing or unstable post-hoc heuristics. Crucially, hallucination patterns exhibit significant sample-level heterogeneity, necessitating adaptive test-time strategies. To address this gap, we introduce a novel test-time hallucination mitigation framework that: (1) constructs multiple low-rank disentangled hallucination subspaces and (2) performs input-adaptive projection into these subspaces, without fine-tuning or compromising generation stability.

### 4.1 CONTRASTIVE DATASET CONSTRUCTION

To extract contrastive signals indicative of hallucinations, we construct a dataset of paired vision-language inputs in the offline phase. Each triplet consists of an image $\mathbf{I}^{(i)}$, a faithful caption $\mathbf{q}^{(i)}$, and a semantically inconsistent (hallucinated) caption $\tilde{\mathbf{q}}^{(i)}$. The hallucinated captions are generated by prompting a language model (e.g., GPT-4) to introduce plausible but absent objects or distort spatial relationships in the original description. Formally, the dataset is defined as:

$$\mathcal{D} = \left\{ \left( \mathbf{I}^{(i)}, \mathbf{q}^{(i)}, \tilde{\mathbf{q}}^{(i)} \right) \right\}_{i=1}^B, \tag{3}$$

where both $\mathbf{q}^{(i)}$ and $\tilde{\mathbf{q}}^{(i)}$ refer to the same image $\mathbf{I}^{(i)}$, but differ in semantic faithfulness.

To analyze the latent representations induced by truthful versus hallucinated captions, we pass each pair through the LVLM and collect hidden states across all layers and positions:

$$\left\{ \mathbf{z}_{\ell,j}^{(i)} \right\}_{\ell,j=1}^{L, J} = f_{\boldsymbol{\theta}}^{\text{LVLM}} \left( \mathbf{I}^{(i)}, \mathbf{q}^{(i)} \right) \tag{4}$$

$$\left\{ \tilde{\mathbf{z}}_{\ell,j}^{(i)} \right\}_{\ell,j=1}^{L, J} = f_{\boldsymbol{\theta}}^{\text{LVLM}} \left( \mathbf{I}^{(i)}, \tilde{\mathbf{q}}^{(i)} \right) \tag{5}$$

Here, $\mathbf{z}_{\ell,j}^{(i)}$ and $\tilde{\mathbf{z}}_{\ell,j}^{(i)}$ represent the hidden state at layer $\ell$ and position $j$, corresponding to the faithful and hallucinated captions, respectively.

**(a)**

**(b)**

Figure 2: Illustration of our test-time hallucination mitigation framework. **(a)** We first construct a contrastive dataset $\mathcal{D}$ to derive a set of low-rank subspaces $\mathbf{V}_r^{(i)}$, $i = 1, \ldots, K$, each capturing a distinct type of hallucination. **(b)** At test time, given an input $\mathbf{I}$, the model adaptively combines the learned subspaces obtained from clustered hallucination-truthful feature differences—into a sample-specific composite subspace $\mathbf{P}$. This subspace is then used to project and modulate internal model activations, effectively suppressing hallucinations without altering the model's parameters.

## 4.2 DISENTANGLED HALLUCINATION SUBSPACES

To uncover semantically distinct directions in the LVLM's activation space that correspond to hallucinations, we perform clustering over feature differences derived from the contrastive dataset $\mathcal{D}$. For each sample $(\mathbf{I}^{(i)}, \mathbf{q}^{(i)}, \tilde{\mathbf{q}}^{(i)}) \in \mathcal{D}$, we compare the model's internal activations under hallucinated and faithful captions.

Given a set of transformer layers $\mathcal{L}$, we first compute token-averaged hidden states from each caption. For every layer $\ell \in \mathcal{L}$, we define:

$$\mathbf{z}_\ell^{(i)} = \frac{1}{J} \sum_{j=1}^{J} \mathbf{z}_{\ell,j}^{(i)}, \quad \tilde{\mathbf{z}}_\ell^{(i)} = \frac{1}{J} \sum_{j=1}^{J} \tilde{\mathbf{z}}_{\ell,j}^{(i)} \tag{6}$$

where $\mathbf{z}_{\ell,j}^{(i)}$ and $\tilde{\mathbf{z}}_{\ell,j}^{(i)}$ are hidden states corresponding to truthful and hallucinated inputs, respectively.

Inspired by Yang et al. (2025), we interpret the deviation between these mean activations as a proxy for semantic drift—a key signal of hallucination. For each layer $\ell$, we collect the representations into matrices:

$$\mathbf{Z}_\ell = [\mathbf{z}_\ell^{(1)}; \ldots; \mathbf{z}_\ell^{(B)}] \in \mathbb{R}^{B \times d} \tag{7}$$

$$\tilde{\mathbf{Z}}_\ell = [\tilde{\mathbf{z}}_\ell^{(1)}; \ldots; \tilde{\mathbf{z}}_\ell^{(B)}] \in \mathbb{R}^{B \times d} \tag{8}$$

We compute layer-wise difference matrices:

$$\mathbf{D}_\ell = \tilde{\mathbf{Z}}_\ell - \mathbf{Z}_\ell \tag{9}$$

To obtain a consolidated representation of hallucination-induced shifts, we average across all considered layers, $\bar{\mathbf{D}} = \frac{1}{|\mathcal{L}|} \sum_{\ell \in \mathcal{L}} \mathbf{D}_\ell$, where each row $\bar{\mathbf{D}}^{(i)} \in \mathbb{R}^d$ now encodes a semantic shift vector for sample $i$. We then apply $K$-means clustering to the rows of $\bar{\mathbf{D}}$, identifying groups of hallucination modes:

$$\bar{\mathbf{D}} = [\bar{\mathbf{D}}^{(1)}; \bar{\mathbf{D}}^{(2)}; \ldots; \bar{\mathbf{D}}^{(K)}] \tag{10}$$

where each cluster $\bar{\mathbf{D}}^{(k)} \in \mathbb{R}^{B_k \times d}$ aggregates samples belonging to hallucination mode $k$.

To extract a compact basis for each hallucination mode, we perform Singular Value Decomposition (SVD) on each cluster:

$$\bar{\mathbf{D}}^{(k)} = \mathbf{U}^{(k)} \mathbf{\Sigma}^{(k)} \mathbf{V}^{(k)\top} \tag{11}$$

The top-$r$ right singular vectors from $\mathbf{V}^{(k)}$, corresponding to the largest singular values, define the basis of the low-rank subspace:

$$\mathbf{V}_r^{(k)} = [\mathbf{v}_{k,1}, \ldots, \mathbf{v}_{k,r}] \in \mathbb{R}^{d \times r} \tag{12}$$

Here, $r$ is a hyperparameter controlling the subspace rank.

Finally, we collect all subspaces into the set $\mathcal{V} = \left\{ \mathbf{V}_r^{(k)} \right\}_{k=1}^{K}$.

Each subspace $\mathbf{V}_r^{(k)}$ captures a distinct semantic mode of hallucination, enabling fine-grained, projection-based suppression. These subspaces are later used to dynamically remove hallucination directions during inference, enhancing the LVLM's robustness under distribution shift.

## 4.3 TEST-TIME HALLUCINATION MITIGATION

Our proposed test-time strategy mitigates hallucinations in LVLMs by dynamically editing the model's internal activations using the subspace set $\mathcal{V}$ constructed in the offline phase. For each test instance, we estimate its hallucination tendency and suppress activation components aligned with hallucination-inducing subspaces.

Given a test image $\mathbf{I} \in \mathbb{R}^{H \times W \times C}$, we first construct a masked variant by zeroing out 70% of its semantically salient regions:

$$\tilde{\mathbf{I}} = \mathbf{I} \odot \mathbf{M}, \tag{13}$$

where $\mathbf{M} \in \{0,1\}^{H \times W \times C}$ is a binary mask and $\odot$ denotes element-wise multiplication.

We then query the LVLM with both the original image $\mathbf{I}$ and the masked image $\tilde{\mathbf{I}}$, using the same textual prompt $\mathbf{q}$. From a set of selected transformer layers $\mathcal{L}$, we extract hidden representations and compute their difference to probe hallucination-sensitive behavior:

$$\boldsymbol{\delta}_{\ell,j} = \frac{1}{J} \sum_{j=1}^{J} \left( \mathbf{z}_{\ell,j}^{(\text{masked})} - \mathbf{z}_{\ell,j}^{(\text{orig})} \right) \in \mathbb{R}^d. \tag{14}$$

Each difference vector $\boldsymbol{\delta}_{\ell,j}$ is projected into the $K$ pre-computed low-rank hallucination subspaces $\{\mathbf{V}_r^{(k)}\}_{k=1}^{K}$, where $\mathbf{V}_r^{(k)} \in \mathbb{R}^{d \times r}$ contains orthonormal basis vectors for hallucination mode $k$. The projection magnitude serves as an alignment score, $s_{k,\ell} = \left\| \boldsymbol{\delta}_{\ell,j}^{\top} \mathbf{V}_r^{(k)} \right\|_2$. These scores are normalized per layer using softmax:

$$\alpha_{k,\ell} = \frac{\exp(s_{k,\ell})}{\sum_{k'=1}^{K} \exp(s_{k',\ell})}. \tag{15}$$

Next, we aggregate alignment scores across layers to obtain a global importance score for each subspace, $\gamma_k = \sum_{\ell \in \mathcal{L}} \alpha_{k,\ell}$. Then, a temperature-controlled softmax is then applied to derive adaptive weights:

$$\beta_k = \frac{\exp(\gamma_k / \tau)}{\sum_{k'=1}^{K} \exp(\gamma_{k'} / \tau)}, \tag{16}$$

where $\tau > 0$ is a temperature hyperparameter. Using these weights, we construct a sample-specific projection matrix:

$$\mathbf{P} = \sum_{k=1}^{K} \beta_k \mathbf{V}_r^{(k)} \mathbf{V}_r^{(k)\top}. \tag{17}$$

Finally, we intervene in the forward pass of the LVLM by editing activations in-place. At each selected layer $\ell \in \mathcal{L}$, token-wise hidden states are projected away from hallucination-prone subspaces:

$$\mathbf{z}_{\ell,j}^{(\text{edited})} = (\mathbf{I}_d - \mathbf{P})\mathbf{z}_{\ell,j}, \quad \text{for } j = 0, 1, \ldots, J - 1. \tag{18}$$

This instance-specific projection dynamically suppresses hallucination-aligned components without requiring any parameter updates, thereby improving factual consistency while preserving the model's generative fluency.

## 5 EXPERIMENTS

We evaluate our proposed method on two established hallucination detection benchmarks—CHAIR Rohrbach et al. (2018) and POPE Li et al. (2023c)—using three representative large vision-language models (LVLMs): LLaVA-1.5 Liu et al. (2023b), MiniGPT-4 Zhu et al. (2023), and mPLUG-Owl2 Ye et al. (2024). We compare our approach against a range of decoding-, tuning-, and projection-based baselines. Comprehensive ablation studies and qualitative analyses are also conducted to assess the impact of each design choice.

| Method | LLaVA-1.5 | | | MiniGPT-4 | | | mPLUG-Owl2 | | |
|---|---|---|---|---|---|---|---|---|---|
| | CHAIR$_S$ ↓ | CHAIR$_I$ ↓ | BLEU↑ | CHAIR$_S$ ↓ | CHAIR$_I$ ↓ | BLEU↑ | CHAIR$_S$ ↓ | CHAIR$_I$ ↓ | BLEU↑ |
| Greedy | $20.40_{\pm2.80}$ | $7.08_{\pm0.33}$ | $15.72_{\pm0.10}$ | $32.40_{\pm2.20}$ | $12.20_{\pm0.42}$ | $14.57_{\pm0.11}$ | $22.90_{\pm0.90}$ | $8.62_{\pm0.11}$ | $15.01_{\pm0.24}$ |
| Beam Search | $19.50_{\pm2.30}$ | $6.84_{\pm0.79}$ | $15.99_{\pm0.14}$ | $30.10_{\pm0.30}$ | $11.87_{\pm0.37}$ | $15.35_{\pm0.24}$ | $20.30_{\pm0.70}$ | $7.62_{\pm0.19}$ | $15.43_{\pm0.05}$ |
| DoLa | $20.20_{\pm2.80}$ | $6.75_{\pm0.54}$ | $15.68_{\pm0.10}$ | $31.90_{\pm3.30}$ | $12.15_{\pm0.89}$ | $14.54_{\pm0.12}$ | $22.40_{\pm1.80}$ | $8.36_{\pm0.04}$ | $15.13_{\pm0.21}$ |
| OPERA | $17.50_{\pm0.50}$ | $6.07_{\pm0.32}$ | $16.02_{\pm0.02}$ | $29.70_{\pm0.30}$ | $11.96_{\pm0.29}$ | $14.82_{\pm0.05}$ | $20.07_{\pm2.07}$ | $7.18_{\pm0.39}$ | $15.41_{\pm0.12}$ |
| VCD | $20.30_{\pm1.10}$ | $7.28_{\pm0.10}$ | $14.53_{\pm0.01}$ | $29.00_{\pm2.80}$ | $12.64_{\pm1.19}$ | $14.42_{\pm0.01}$ | $22.80_{\pm0.80}$ | $8.68_{\pm0.17}$ | $15.14_{\pm0.13}$ |
| Woodpecker | $23.85_{\pm4.62}$ | $7.50_{\pm0.01}$ | $17.05_{\pm0.00}$ | $28.87_{\pm2.20}$ | $10.20_{\pm0.85}$ | $15.30_{\pm0.01}$ | $26.33_{\pm1.98}$ | $8.43_{\pm0.80}$ | $16.43_{\pm0.00}$ |
| LURE | $19.48_{\pm2.35}$ | $6.50_{\pm0.38}$ | $15.97_{\pm0.01}$ | $27.88_{\pm2.25}$ | $10.20_{\pm0.85}$ | $15.03_{\pm0.01}$ | $21.27_{\pm0.06}$ | $7.67_{\pm0.16}$ | $15.65_{\pm0.15}$ |
| HALC | $16.90_{\pm2.10}$ | $5.72_{\pm0.55}$ | $16.02_{\pm0.04}$ | $25.20_{\pm2.00}$ | $9.42_{\pm0.41}$ | $14.91_{\pm0.13}$ | $18.80_{\pm1.20}$ | $7.00_{\pm0.01}$ | $15.33_{\pm0.24}$ |
| Nullu | $15.20_{\pm0.60}$ | $5.30_{\pm0.03}$ | $15.69_{\pm0.04}$ | $21.40_{\pm1.00}$ | $8.99_{\pm0.36}$ | $14.81_{\pm0.06}$ | $15.60_{\pm1.20}$ | $5.77_{\pm0.01}$ | $15.45_{\pm0.01}$ |
| **Ours** | $\mathbf{14.60}_{\pm0.35}$ | $\mathbf{4.92}_{\pm0.05}$ | $15.63_{\pm0.01}$ | $\mathbf{21.01}_{\pm0.95}$ | $\mathbf{8.64}_{\pm0.22}$ | $14.87_{\pm0.03}$ | $\mathbf{15.21}_{\pm1.01}$ | $\mathbf{5.46}_{\pm0.01}$ | $15.69_{\pm0.02}$ |

Table 1: Comparison of different methods on CHAIR$_S$, CHAIR$_I$, and BLEU metrics across LLaVA-1.5, MiniGPT-4, and mPLUG-Owl2.

### 5.1 DATASETS

We evaluate our method on four benchmark datasets widely used to assess hallucination and visual grounding in large vision-language models: CHAIR Rohrbach et al. (2018) and POPE Li et al. (2023c). These benchmarks collectively examine both object-level hallucination and multimodal consistency across image captioning and visual question answering tasks.

**CHAIR.** CHAIR focuses on object hallucination in image captions by verifying whether the mentioned objects are visually grounded in the image. It reports two metrics: CHAIR$_S$, the percentage of captions containing hallucinated objects, and CHAIR$_I$, the proportion of hallucinated object tokens among all generated object mentions—lower values indicate better grounding. We also report BLEU to assess caption fluency. Following standard protocol Yang et al. (2025), we prompt each model with: *"Please describe this image in detail."*

**POPE.** POPE evaluates hallucination through yes/no questions about object presence in images. It comprises three query types—*random*, *frequent*, and *adversarial*—to probe model robustness under varying difficulty levels. In addition, we adopt the Offline POPE (OPOPE) variant Li et al. (2023c), which analyzes hallucination by checking whether non-existent objects appear in generated captions, rather than in direct answers to object queries.

### 5.2 BASELINES AND EVALUATION SETUP

We compare our method against a diverse set of recent hallucination mitigation approaches, including decoding-based, projection-based, and tuning-based techniques. Specifically, we evaluate against DoLa Chuang et al. (2023), OPERA Huang et al. (2024), VCD Leng et al. (2024a), Woodpecker Yin et al. (2024), LURE Zhou et al. (2023), HALC Chen et al. (2024), and Nullu Yang et al. (2025). We also include standard decoding strategies (*Greedy*, *Beam Search*) and tuning-based baselines such as LURE and MiniGPT-4 fine-tuned variants.

Our method is applied to three strong LVLM backbones: LLaVA-1.5 Liu et al. (2023b), MiniGPT-4 Zhu et al. (2023), and mPLUG-Owl2 Ye et al. (2024), all evaluated without any model fine-tuning. Following prior work Yang et al. (2025), we treat hallucination mitigation as a test-time operation and use consistent prompts across models for fair comparison. While our approach builds upon Nullu Yang et al. (2025), it differs by learning multiple HalluSpaces through clustering and applying adaptive, sample-specific null space projections.

| Method | LLaVA-1.5 | | | MiniGPT-4 | | | mPLUG-Owl2 | | |
|---|---|---|---|---|---|---|---|---|---|
| | Accuracy↑ | Precision↑ | F score↑ | Accuracy↑ | Precision↑ | F score↑ | Accuracy↑ | Precision↑ | F score↑ |
| Greedy | $79.14_{\pm0.89}$ | $91.98_{\pm0.82}$ | $90.45_{\pm0.86}$ | $71.22_{\pm1.27}$ | $93.72_{\pm1.02}$ | $90.04_{\pm1.23}$ | $76.46_{\pm0.92}$ | $88.85_{\pm1.15}$ | $87.29_{\pm1.15}$ |
| Beam Search | $79.41_{\pm0.69}$ | $92.52_{\pm0.55}$ | $90.96_{\pm0.59}$ | $71.65_{\pm1.15}$ | $94.70_{\pm0.60}$ | $90.97_{\pm0.85}$ | $76.76_{\pm1.02}$ | $90.28_{\pm0.80}$ | $88.56_{\pm0.87}$ |
| DoLa | $78.98_{\pm0.56}$ | $91.66_{\pm0.81}$ | $90.15_{\pm0.79}$ | $71.28_{\pm1.15}$ | $93.92_{\pm0.83}$ | $90.22_{\pm1.04}$ | $76.07_{\pm1.09}$ | $88.54_{\pm1.25}$ | $86.95_{\pm1.27}$ |
| OPERA | $79.29_{\pm0.32}$ | $92.25_{\pm0.07}$ | $90.71_{\pm0.11}$ | $70.48_{\pm1.63}$ | $94.41_{\pm1.11}$ | $90.66_{\pm1.42}$ | $75.49_{\pm1.29}$ | $91.23_{\pm1.06}$ | $89.11_{\pm1.17}$ |
| VCD | $78.01_{\pm0.75}$ | $91.33_{\pm0.88}$ | $89.69_{\pm0.89}$ | $70.83_{\pm1.83}$ | $92.31_{\pm0.88}$ | $88.76_{\pm1.29}$ | $75.49_{\pm1.27}$ | $88.75_{\pm1.56}$ | $87.02_{\pm1.57}$ |
| HALC | $77.87_{\pm0.22}$ | $93.17_{\pm0.39}$ | $91.25_{\pm0.38}$ | $71.17_{\pm0.89}$ | $94.88_{\pm0.15}$ | $90.95_{\pm0.42}$ | $74.93_{\pm1.09}$ | $90.20_{\pm0.90}$ | $88.12_{\pm0.99}$ |
| Nullu | $79.52_{\pm0.04}$ | $93.46_{\pm0.03}$ | $91.79_{\pm0.04}$ | $71.92_{\pm0.39}$ | $95.96_{\pm0.65}$ | $92.07_{\pm0.65}$ | $77.09_{\pm1.37}$ | $92.83_{\pm0.29}$ | $90.80_{\pm0.52}$ |
| **Ours** | $\mathbf{79.80}_{\pm0.02}$ | $\mathbf{93.6}_{\pm0.02}$ | $\mathbf{91.92}_{\pm0.06}$ | $\mathbf{72.2}_{\pm0.33}$ | $\mathbf{96.02}_{\pm0.62}$ | $\mathbf{92.32}_{\pm0.32}$ | $\mathbf{78.12}_{\pm1.02}$ | $\mathbf{93.4}_{\pm0.22}$ | $\mathbf{91.68}_{\pm0.62}$ |

Table 2: The OPOPE evaluation results on MSCOCO dataset of LVLMs with different methods for mitigating OH. Higher accuracy, precision, and F score indicate better performance.

## 5.3 IMPLEMENTATION DETAILS

We apply our method without any fine-tuning to three pretrained LVLM backbones: LLaVA-1.5 Liu et al. (2023b), MiniGPT-4 Zhu et al. (2023), and mPLUG-Owl2 Ye et al. (2024). For each model and benchmark, we vary the number of HalluSpace clusters and the dimensionality of the projection bases. On the CHAIR benchmark, we use 5 clusters and 32 basis vectors for mPLUG-Owl2, and 11 clusters with 8 basis vectors for MiniGPT-4. On POPE, we use 6 clusters for LLaVA-1.5 and 11 clusters for MiniGPT-4. These settings were selected based on preliminary experiments balancing performance and computational efficiency. All evaluations are conducted on the MSCOCO validation split, and we report the mean and standard deviation across ten independent runs to account for variance introduced by clustering.

## 5.4 RESULTS ON CHAIR

Table 1 presents results on the CHAIR benchmark, using $CHAIR_S$, $CHAIR_I$, and BLEU as metrics. Across all three base models, our method consistently reduces both sentence-level and image-level hallucination rates compared to all baselines, including strong constrained decoding methods (HALC, DoLa) and null-space projection (Nullu). On LLaVA-1.5, we reduce $CHAIR_I$ from 5.30 (Nullu) to 4.92, while maintaining comparable BLEU. The trend holds for MiniGPT-4 and mPLUG-Owl2, where our method either matches or slightly improves BLEU, while reducing hallucination errors. This demonstrates that our adaptive, sample-specific projection reduces hallucination rates more effectively than all baselines, while maintaining competitive BLEU scores.

## 5.5 RESULTS ON POPE

Table 2 reports results on POPE, which evaluates factual alignment in a question-answering setting. Our method achieves the best performance across all three metrics on mPLUG-Owl2, improving F-score from 90.80 (Nullu) to 91.60, along with consistent gains in accuracy and precision. For LLaVA-1.5 and MiniGPT-4, our method yields slight improvements in accuracy but does not surpass Nullu in F-score or precision. These results suggest that our sample-specific suppression mechanism is particularly effective for stronger base models such as mPLUG-Owl2, where it enhances factual alignment without compromising fluency. Moreover, the improvements demonstrate the adaptability of our framework, showing that even modest gains can accumulate to meaningful robustness in challenging benchmarks.

## 5.6 ABLATION STUDY

We evaluate the impact of cluster count and basis dimensionality on hallucination mitigation. Fewer clusters result in reduced specificity, while smaller bases fail to isolate fine-grained spurious features. Our method outperforms Nullu even with fewer clusters, due to per-sample adaptivity.

**Number of subspaces:** In this experiment (see Figure 3 **(left)**), we evaluate the effect of the number of subspaces on the performance of the LLaVA model. The optimal number of subspaces for each model is first determined on the COCO training set and then applied to the test set. As shown in

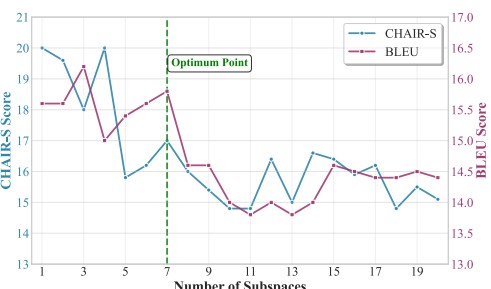 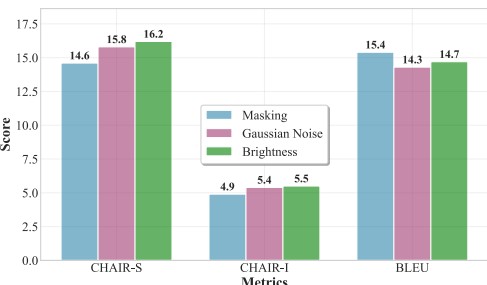

Figure 3: **(left)** The impact of the number of subspaces in the LLaVA-1.5 model within our proposed method. **(right)** We assess different perturbation strategies—masking, Gaussian, and blurring.

| Metric | Number of basis | | | | | | |
|---|---|---|---|---|---|---|---|
| | 1 | 2 | 4 | 8 | 16 | 32 | 64 |
| **CHAIR$_S$ ↓** | 16.2 | 15.2 | 14.6 | 14.5 | 13.2 | 12.5 | 11.3 |
| **CHAIR$_I$ ↓** | 6.1 | 5.3 | 4.9 | 4.8 | 4.2 | 4.0 | 3.9 |
| **BLEU ↑** | 15.6 | 15.5 | 15.6 | 14.2 | 13.4 | 12.9 | 12.7 |

Table 3: The influence of the number of subspaces in our method applied to the LLaVA-1.5 model.

the figure, we select 7 subspaces for LLaVA, as this configuration minimizes CHAIR$_S$ score while maximizing the BLEU score. Similarly, we choose 11 subspaces for MiniGPT-4 and 5 for mPLUG-Owl2 based on the same optimization criteria. These findings highlight that the optimal number of subspaces varies across models, underscoring the importance of tailoring the editing strategy to each LVLM. Moreover, the results confirm that increasing the number of subspaces beyond the optimal point does not necessarily yield further improvements and may even degrade performance.

**Influence of image perturbations:** We evaluate the impact of different perturbation strategies—masking, Gaussian noise, and blurring—applied to test samples during test time in our hallucination mitigation method (see Figure 3 **(right)**). As shown in the figure, the masking strategy yields the best performance. This suggests that masking provides a more effective signal for disentangling hallucination-prone directions compared to other perturbations. Overall, the results highlight the importance of carefully selecting perturbation strategies when designing adaptive editing methods.

**Number of basis vectors:** This experiment (see Table 3) evaluates the impact of the number of basis vectors in the LLaVA. As the number of basis vectors increases beyond 4, we observe a decrease in CHAIR$_S$ and CHAIR$_I$ scores, while the BLEU score decreases. This trend is undesirable, as a lower BLEU score in this context indicates that the LVLM model is generating responses that are less grounded in the image content. As a result, we choose the number of basis vectors 4 for the LLaVA model.

## 6  CONCLUSION

We have presented a novel training-free, test-time adaptation method for mitigating hallucinations in large vision-language models. By modeling hallucinations with multiple low-rank subspaces derived from clustered hallucination-truthful feature pairs, our approach captures the diverse and sample-specific nature of hallucination patterns. Unlike existing methods, our framework dynamically adapts the hallucination suppression subspace for each test input, allowing for fine-grained, input-dependent corrections without permanently altering the model. Our extensive evaluation on six benchmarks and across four LVLM families confirms that our method consistently improves hallucination robustness while maintaining the original model's integrity. This work offers a practical and effective solution toward more reliable LVLM deployments in real-world applications.

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
