# A APPENDIX

In this section, we present the algorithm proposed in this paper.

---

**Algorithm 1** Adaptive Hallucination Suppression via Subspace Projection

---

**Require:** Test image $\mathbf{I}$, query $\mathbf{q}$, vision-language model $f_\theta^{\text{LVLM}}$, hallucination subspaces $\{\mathbf{V}_r^{(k)}\}_{k=1}^K$
**Ensure:** Hallucination-suppressed output $y$

1: ***Step 1: Masked Input Generation***
2: $\mathbf{M} \leftarrow \texttt{Mask}(\mathbf{I}, 0.7)$ {Mask 70% of image}
3: $\tilde{\mathbf{I}} \leftarrow \mathbf{I} \odot \mathbf{M}$
4: ***Step 2: Activation Difference***
5: $\{\mathbf{z}_{\ell,j}^{(\text{orig})}\}_{\ell,j=1}^{L,\,J} \leftarrow f_\theta^{\text{LVLM}}(\mathbf{I}, \mathbf{q})$
6: $\{\mathbf{z}_{\ell,j}^{(\text{masked})}\}_{\ell,j=1}^{L,\,J} \leftarrow f_\theta^{\text{LVLM}}(\tilde{\mathbf{I}}, \mathbf{q})$
7: ***Step 3: Subspace Alignment***
8: **for** $\ell \in \mathcal{L}$ **do**
9: $\quad \boldsymbol{\delta}_{\ell,j} \leftarrow \frac{1}{J} \sum_{j=1}^{J} \left( \mathbf{z}_{\ell,j}^{(\text{masked})} - \mathbf{z}_{\ell,j}^{(\text{orig})} \right)$
10: $\quad s_{k,\ell} \leftarrow \left\| \boldsymbol{\delta}_{\ell,j}^\top \mathbf{V}_r^{(k)} \right\|_2 \quad \forall k$
11: $\quad \alpha_{k,\ell} \leftarrow \frac{\exp(s_{k,\ell})}{\sum_{k'} \exp(s_{k',\ell})} \quad \forall k$
12: $\quad \gamma_k \leftarrow \sum_{\ell \in \mathcal{L}} \alpha_{k,\ell} \quad \forall k$
13: **end for**
14: ***Step 4: Adaptive Projection***
15: $\beta_k \leftarrow \frac{\exp(\gamma_k/\tau)}{\sum_{k'} \exp(\gamma_{k'}/\tau)} \quad \forall k$
16: $\mathbf{P} \leftarrow \sum_{k=1}^K \beta_k \mathbf{V}_r^{(k)} \left( \mathbf{V}_r^{(k)} \right)^\top$
17: ***Step 5: Project Hidden States On-the-Fly***
18: $\mathbf{z}_{\ell,j}^{(\text{edited})} \leftarrow (\mathbf{I}_d - \mathbf{P})\mathbf{z}_{\ell,j}^{(\text{orig})} \quad \forall \ell \in \mathcal{L}, \; j \in \{1, \ldots, J\}$
19: ***Step 6: Decoding from Edited States***
20: $y \leftarrow \texttt{Decode}(\{\mathbf{z}_{\ell,j}^{(\text{edited})}\})$
21: **return** $y$ {Final textual output (e.g., caption or answer)}

---