# OpenReview forum: "Hallucination Mitigation in Large Vision-Language Models via Adaptive Multi-Subspace Projection"
_ICLR.cc/2026/Conference — ICLR 2026 Conference Withdrawn Submission_

### Official Review · Reviewer_c3Mr · 2025-10-31

**Soundness:** 4
**Presentation:** 3
**Contribution:** 3
**Rating:** 6
**Confidence:** 3

**Summary:**

This paper proposes a training-free method to mitigate hallucinations in Large Vision-Language Models (LVLMs) through **adaptive multi-subspace projection**. The authors argue that existing model-editing approaches like Nullu use a single global subspace that fails to capture diverse hallucination patterns across different inputs. So, they construct multiple disentangled hallucination subspaces via K-means clustering and SVD, then adaptively weight these subspaces at test time based on input-specific hallucination signals derived from masked image perturbations. The method is evaluated on CHAIR and POPE benchmarks across three LVLM families (LLaVA-1.5, MiniGPT-4, mPLUG-Owl2), showing improvements over existing baselines including the recent Nullu method.

**Strengths:**

1. The paper identifies a limitation of existing fixed model-editing methods: a single global subspace cannot adapt to the heterogeneous hallucination patterns that vary across different inputs. In my view, this observation is insightful and the proposed solution of using multiple subspaces with adaptive weighting represents a natural and promising direction for improvement.

2. The inclusion of ablation studies on the number of subspaces, basis dimensions, and perturbation strategies demonstrates investigation of the method's behavior. The consistency of improvements across different settings is encouraging.

3. The method maintains the training-free property which is valuable for practical deployment. Unlike fine-tuning approaches that require curated datasets and substantial computational resources, the proposed approach offers a reasonable middle ground by preprocessing subspaces offline and applying lightweight adaptive weighting at test time.

4. The two-stage framework is well-designed and the mathematical formulation is generally clear.

**Weaknesses:**

- In my opinion, the writing of the paper could be improved. The reported improvements over Nullu are relatively modest in limited statistical validation of improvements, and given the standard deviations shown some gains may not be significant. The paper would be much stronger with paired t-tests or similar statistical validation to confirm these improvements are reliable rather than random variation.

-  While the paper claims efficiency advantages, no actual inference times, memory usage, or FLOPs are reported to validate this. Additionally, several key technical details lack clarity: **semantically salient regions** for masking (see in Equation 13) is never defined.

- Table 3 reveals that increasing basis vectors improves hallucination metrics but causes substantial BLEU degradation, suggesting over-suppression of legitimate content. While the authors select **balanced** hyperparameters empirically, there is no principled guidance for navigating this trade-off in new settings, and no theoretical understanding of why it occurs.

**Questions:**

1. Could you add statistical significance tests to validate that the improvements over Nullu are reliable rather than within-noise variation? This would substantially strengthen the empirical claims.

2. How are **semantically salient regions** identified for the masking operation? Please provide implementation details or point to the specific saliency method used.

---

### Official Review · Reviewer_ftf2 · 2025-10-31

**Soundness:** 2
**Presentation:** 2
**Contribution:** 2
**Rating:** 2
**Confidence:** 4

**Summary:**

This paper proposes a training-free framework to mitigate hallucinations in LVLMs. The method first constructs a set of disentangled hallucination subspaces via SVD and K-means. Then, at inference time, the model adaptively creates specific weights from the subspaces to alleviate the hallucination, via two forward queries of the original image and masked image.

**Strengths:**

1. The novelty is sound. The authors find an adaptive method to calculate specific weights for different queries, which is often neglected in other hallucination papers, as the hallucination type is different for different inputs.
2. The general method builds up with the training-free methods, while adaptively creating specific weights from a clustered subspace, which seems to be superior to some other training-free methods.
3. The method is well evaluated across different base models.

**Weaknesses:**

1. The paper's Table 3 shows that as hallucination suppression increases (using more basis vectors), the BLEU score drops significantly. While BLEU is not a comprehensive metric for modern LVLMs, this still raises concerns about the degradation in general model performance. The evaluation is narrow on hallucination benchmarks and lacks testing on broader, general-purpose benchmarks (MMMU/VQAv2/...) to confirm that the model's abilities are not so compromised.

2. One of the core claims is that it identifies distinct hallucination modes. However, the authors do not provide any qualitative analysis or evidence, or visualization to validate that these disentangled subspaces actually correspond to semantically different types of hallucinations.

3. The method requires two forward passes at test time (one for the original image and one for the masked one) to get a diff. This will introduce extra computation. Moreover, it remains unclear why the specific difference signal serves as a proxy for the input-specific hallucination signal. Lastly, the authors seem to miss the exact strategy for masking. Is it random black masking or other strategies?

**Questions:**

N/A

---

### Official Review · Reviewer_Wtxx · 2025-11-01

**Soundness:** 2
**Presentation:** 3
**Contribution:** 2
**Rating:** 4
**Confidence:** 4

**Summary:**

This paper proposes a training-free way to reduce hallucinations in large vision-language models (LVLMs) by editing their internal activations at test time instead of fine-tuning them. The key idea is to build multiple low-rank “hallucination subspaces,” each representing a different type of hallucination, by comparing model states from truthful vs. hallucinated captions. At inference, the model estimates which hallucination modes are most likely for the current input image, then dynamically projects its hidden representations away from those directions, suppressing ungrounded content while keeping image-relevant semantics. Experiments on benchmarks like CHAIR and POPE across models such as LLaVA-1.5, MiniGPT-4, and mPLUG-Owl2 show that this adaptive multi-subspace projection reduces hallucinations more consistently than prior decoding-based or single-subspace editing methods.

**Strengths:**

1. The paper models hallucination not as one global direction but as multiple disentangled subspaces, each tied to a different hallucination mode. At test time it adaptively weighs these subspaces for the current input and projects away the most risky directions, which leads to stronger hallucination suppression than fixed one-subspace editing.

2. The ablation shows that different LVLM backbones prefer different numbers of subspaces (e.g., 7 for LLaVA-1.5, 11 for MiniGPT-4, 5 for mPLUG-Owl2). This suggests each model has its own “hallucination landscape,” rather than a universal structure. Making these subspaces interpretable in semantic terms (e.g., “spurious object insertion,” “wrong spatial relation”) would be a valuable next step.

**Weaknesses:**

1. The method needs two forward passes at inference: it runs the LVLM on both the original image and a perturbed/masked version to estimate which hallucination modes are likely, then applies the adaptive projection. Prior fixed-edit approaches only require a single edited forward. Authors need to report a compute/runtime comparison against those baselines.

2. The “contrastive dataset” used to build the hallucination subspaces is under-specified. The paper does not state where the images/captions come from, how large this dataset is. Without dataset source/scale, it is hard to judge fairness and reproducibility.

3. The experiments are only on older/open LVLMs (LLaVA-1.5, MiniGPT-4, mPLUG-Owl2). There is no evidence that the approach still works on newer high-capability MLLMs (e.g., recent Qwen2.5-VL–series) .

**Questions:**

Please refer to the weaknesses part.

---

### Note · Authors · 2025-11-14

I have read and agree with the venue's withdrawal policy on behalf of myself and my co-authors.